# An In Situ Formation of Ionic Liquid for Enrichment of Triazole Fungicides in Food Applications Followed by HPLC Determination

**DOI:** 10.3390/molecules27113416

**Published:** 2022-05-25

**Authors:** Rawikan Kachangoon, Jitlada Vichapong, Yanawath Santaladchaiyakit, Supalax Srijaranai

**Affiliations:** 1Creative Chemistry and Innovation Research Unit, Department of Chemistry and Center of Excellent for Innovation in Chemistry, Faculty of Science, Mahasarakham University, Maha Sarakham 44150, Thailand; r.kachangoon@hotmail.com; 2Multidisplinary Research Unit of Pure and Applied Chemistry (MRUPAC), Department of Chemistry and Center of Excellent for Innovation in Chemistry, Faculty of Science, Mahasarakham University, Maha Sarakham 44150, Thailand; 3Department of Chemistry, Faculty of Engineering, Rajamangala University of Technology Isan, Khon Kaen Campus, Khon Kaen 40000, Thailand; sanyanawa@gmail.com; 4Materials Chemistry Research Center, Department of Chemistry and Center of Excellent for Innovation in Chemistry, Faculty of Science, Khon Kaen University, Khon Kaen 40002, Thailand; supalax@kku.ac.th

**Keywords:** ionic liquid, in situ extraction, triazole fungicides, extraction, HPLC

## Abstract

An in situ formation of ionic liquid was used for preconcentration of four triazole fungicides in food samples. The microextraction method was used for the first time in the literature for preconcentration of triazole fungicides. In the developed method, tributylhexadecylphosphonium bromide ([P_44412_]Br) and potassium hexafluorophosphate (KPF_6_) were used for the formation of hydrophobic ionic liquid. After centrifugation, the fine microdroplets were produced in one step, providing the extraction step in a quick and environmentally friendly manner. The functional group of the hydrophobic ionic liquid was investigated using FT-IR. Various extraction parameters were studied and optimized. In the extraction method, 0.01 g of [P_44412_]Br and 0.01 g of KPF_6_, centrifugation at 4500 rpm for 10 min were used. The optimized technique provided a good linear range (90–1000 μg L^−1^) and high extraction recovery, with a low limit of detection (30–50 μg L^−1^). Methods for the proposed in situ formation of ionic liquid were successfully applied to honey, fruit juice, and egg matrices. The recoveries were obtained in a satisfactory range of 62–112%. The results confirmed the suitability of the proposed microextraction method for selective extraction and quantification of triazole fungicides.

## 1. Introduction

Triazole fungicides (TFs) are a class of extremely efficient systemic fungicides that contain a hydroxyl group (ketone group), a substituted phenyl group, and a 1,2,4-triazole group in the main chain [1]. The TFs are promoted with broad-spectrum, internal absorption characteristics, providing good stability for a long time [2]. They are commonly applied in the cultivation and postharvest storage of crops to protect them from fungal infections [3,4]. However, they may disturb endocrine activities, and have induced developmental toxicity in animals and human health [5]. For this reason, a reliable and effective method is still needed for the determination of triazole fungicides in various matrices.

Chromatographic methods such as liquid chromatography [6,7,8,9,10] and gas chromatography [11,12,13,14,15] have been used for the simultaneous analysis of triazole fungicides. Due to the trace levels of triazole fungicides in complex matrices, some preconcentration/extraction procedures are necessary prior to instrumental analysis. The advancement of sample-preparation methods has mainly promoted their simplification, miniaturization (reduction in the amounts of sample and extraction solvent), and environmental friendliness (use of an alternative extraction solvent) [4]. Recently, miniaturized techniques have been advanced to enrich triazoles in various matrices, such as ultrasonic-assisted dispersive solid-phase extraction (UA-DSPE) [16], magnetic solid-phase extraction (MSPE) [6], homogeneous liquid–liquid extraction based on salting out (SHLLE) [14], vortex-assisted ionic liquid dispersive liquid–liquid microextraction (VA-IL-DLLME) [9], aqueous two-phase extraction (ATPE) [17], and microsolid-phase extraction [18].

Room temperature ionic liquids (RTILs), which are defined as ‘‘alternative’’ solvents, are promising and suitable solvents that have been recently used as replacement extraction solvents for sample preparation and preconcentration methods [19]. They have some unique attributes, such as a negligible vapor pressure, high thermal stability, good extractability for various organic compounds and metal ions as neutral or charged complexes, and tunable viscosity and miscibility with water and organic solvents [20]. The RTIL types are alkylammonium, tetraalkylammonium, tetraalkylphosphonium, 1,3-dialkylimidazolium, and N-alkylpyridinium salts formed with weak nucleophilic anions such as hexafluorophosphate, tetrafluoroborate, and perfluoroalkylsulfonate [21]. There are some reports on sample preconcentration using RTILs [19,20,21,22]. Some of extraction methods were based on liquid–liquid microextraction and dispersive liquid–liquid microextraction (DLLME). Generally in the DLLME process, a disperser solvent is required to induce the mass transfer of the extraction solvents into an aqueous solution [23]. In situ DLLME was developed to avoid the use of these dispersive solvents [24,25]. In this method, a hydrophilic RTIL was used as a starting extraction solvent capable of completely dissolving in the aqueous analyte solution. Then, an in situ metathesis reaction was used to form an RTIL immiscible with water by adding an ion-exchange reagent into the aqueous solution to complete the extraction process. This exchange process formed a cloudy solution with fine microdroplets, which exceedingly improved the surface area of the RTIL, resulting in the high extraction efficiency of the target analytes. Due to the high price of RTILs, the metathesis reaction and extraction were investigated in a single step, making the transfer of analyte into the extracting phase very quick and efficient. The benefits of the in situ metathesis reaction-assisted IL were the rejection of the requirement for using a disperser solvent and a remarkable increase in the surface area of the IL extraction solvent.

In this study, an in situ metathesis reaction that generated an ionic liquid was combined with a liquid–liquid microextraction for the preconcentration of triazole fungicides prior to their analysis using high-performance liquid chromatography. A tributyldodecylphosphonium bromide ([P_44412_]Br) as the starting extraction solvent and an ion-exchange reagent (KPF_6_) were used to produce a hydrophobic ionic liquid (IL) ([P_44412_][PF_6_]). The starting extraction solvent could be completely dissolved in the aqueous solution, which promoted analyte extraction in the absence of a dispersion solvent. In this work, an in situ metathesis reaction was performed to yield a cloudy solution with fine microdroplets, which significantly increased the extraction efficiency. After centrifugation, the extraction solvent was easily collected with a syringe after solidification. Using in situ formation, the fine microdroplets were produced in one step, providing a short extraction time and environmental friendliness. Various experimental parameters that affected the extraction efficiency of the method were investigated and optimized. Finally, the selected extraction conditions were used to determine triazole fungicide residues in water, honey, fruit juice, and egg matrices.

## 2. Results and Discussion

### 2.1. Characterization of In Situ Ionic Liquid

The halide anion of tributylhexadecylphosphonium bromide ([P_44412_]Br) and potassium hexafluorophosphate (KPF_6_) was the main force for the formation of the hydrophobic ionic liquid. FT-IR spectra were used to confirm the formation of hydrogen bonding, as shown in Figure 1. In the FT-IR spectra, the characteristic peaks presented at 2850, 2920, and 2960 cm^−1^ were assigned to the C-H stretching or CH_3_ stretching, while those at 1410 and 1465 cm^−1^ were attributed to the C-H bending and CH_3_ bending vibrations of pure [P_44412_]Br [26]. Moreover, the FT-IR spectra of the characteristic peaks showed the halogen compound vibrations at 577 and 795 cm^−1^. This may have been due to a transfer of the bromide ion cloud electron to hydrogen bonding, and consequently, a decrease in the force constant [27]. Thus, the shift of the hexafluorophosphate counter-ion vibrations suggested the existence of hydrogen bonding between [P_44412_]Br and KPF_6_ when the hydrophobic ionic liquid was formed.

### 2.2. Optimization of In Situ Metathesis-Reaction-Generated Ionic Liquid Combined with Liquid–Liquid Microextraction

Due to their low concentrations and matrix interferences in real samples, it is difficult to directly analyze triazole fungicides. Therefore, a sample-preparation method is necessary before analysis. In this work, an in situ metathesis reaction that generated the ionic liquid was combined with a liquid–liquid microextraction in the triazole fungicide analysis. In order to obtain a high extraction efficiency, various parameters had to be investigated. The optimization was carried out using the aqueous solution (10 mL) containing 100 μg L^−1^ of each triazole fungicides. All experiments were performed at least in triplicate.

In order to form the ionic liquid ([P_44412_][PF_6_]), [P_44412_]Br and KPF_6_ were selected because the starting extraction solvent could be completely dissolved in the aqueous solution, which promoted the analyte extraction in the absence of a dispersion solvent; the melting of this IL ([P_44412_][PF_6_]) was between 10 and 30 °C, and it could solidify at low temperatures [24]. Moreover, [P_44412_][PF_6_] is denser than water, therefore it was easy to collect as the bottom layer after centrifugation. The amount of [P_44412_]Br was studied in the range of 0.01–0.10 g. The results are shown in Figure 2. The results showed that the maximum peak area was obtained at 0.01 g of [P_44412_]Br. Therefore, 0.01 g of [P_44412_]Br was chosen.

The amount of KPF_6_ varied in a range of 0.01–0.10 g. The results are shown in Figure 3. The results showed that the maximum peak area was obtained at 0.01 g of KPF_6_. Then, the signal decreased at a higher amount of KPF_6._ Thus, 0.01 g of KPF_6_ was selected. The molar ratio of [P_44412_]Br to KPF_6_ was selected as 1:2.5 because the excess KPF_6_ ensured 100% completion of the in situ metathesis reaction [24].

Because the process of mass transfer and completely phase separation in an extraction procedure should be time-dependent, the effects of extraction speed and time on the peak area was studied [22]. The effect of centrifugation speed was evaluated in the range of 1500–5000 rpm (see Figure 4). It was found that the peak areas of all the analytes increased up to a 4500 rpm centrifugation speed, above which the peak areas slightly decreased due to disintegration of the phase of target analytes. Therefore, a centrifugation speed of 4500 rpm was selected. The extraction times studied ranged from 3 to 15 min, while other experimental conditions were kept constant (data not shown). The peak areas of most of the triazoles increased with increases in the extraction time, and reached the highest at 10 min. Therefore, 10 min was chosen to ensure an efficient extraction.

Before HPLC analysis, acetonitrile (ACN) was added to dissolve the RTIL-rich phase due to its solubility property in the [P_44412_]PF_6_ phase and the compatibility with the mobile phase being used. The ACN volume was investigated in the range of 30–250 μL (as shown in Figure 5). It was found that ACN 200 μL provided the highest peak areas of all analytes. After that, the peak area decreased due to the dilution effect. When using ACN at less than 30 μL, the phase could not be completely dissolved. Thus, 200 μL of ACN was chosen. 

### 2.3. Analytical Performance of the Proposed Method

Under the chosen extraction condition, a series of experiment were conducted to study the enrichment factors (EFs), extraction recoveries (ERs), linear ranges, limits of detection (LODs), limits of quantitation (LOQs), and precisions (intraday and interday). The results are shown in Table 1. The calibration graphs were linear over concentration ranges of 90–1000 µg L^−1^ for myclobutanil, triadimefon, and tebuconazole, and 150–1000 µg L^−1^ for hexaconazole, with a coefficient of determination in the range of 0.9988–0.9991. The LODs were evaluated as the concentrations giving a signal-to-noise ratio of 3 (S/N = 3), and were in the range of 30–50 μg L^−1^. The LOQs (S/N = 10) were in the range of 90–150 μg L^−1^. The relative standard deviation (RSD) was determined using five solutions of 150 µg L^−1^ of each triazole fungicide. The RSD values of the retention times and peak areas were in the ranges of 0.40–0.80% and 4.84–6.23%, respectively. The EFs were in the range of 8.53–12.26. The ERs were in the range of 59.71–85.82. The chromatograms obtained from the proposed preconcentration method and direct analysis (without preconcentration) are shown in Figure 6. It was found that the developed method showed high chromatographic signals when compared to the direct analysis.

### 2.4. Analysis of Real Samples

To study the applicability of the proposed method, an in situ formation and preconcentration method using the ionic liquid was applied in the determination of triazole fungicides in honey, fruit juice, and egg samples. A standard addition method was applied to study the matrix effect (ME). ME(%) is expressed as the ratio of the slopes obtained from calibration curves of each analyte spiked into the samples to those obtained after extraction using the proposed method, according to Equation (1) [28]: (1)$$\mathrm{ME}\;(\%)=\frac{\mathrm{slope}\;\mathrm{of}\;\mathrm{spiked}\;\mathrm{real}\;\mathrm{sample}}{\mathrm{slope}\;\mathrm{of}\;\mathrm{standard}\;\mathrm{solution}}\times 100$$

In the absence of the ME, the slope of the calibration curves from both the standard solution and spiked real samples should be similar (ME (%) ≈ 100%). However, in the presence of ME, the signal intensity for the analytes can decrease or increase. Generally, ME values between 80 and 120% indicate no ME, ME values between 50 and 80% or 120 and 150% indicate minor MEs, and ME values less than 50% or greater than 150% indicate major MEs [29]. It was found that the MEs (%) were in the range of 76.9–113.9%.

In order to confirm the accuracy of the proposed method, the relative recoveries (RRs) were investigated using an analysis of three real samples spiked with five triazole fungicides at a concentration of 150 μg·L^−1^ within one day. As shown in Table 2, acceptable recoveries (62–112%) with relative standard deviations (RSDs) of less than 7.9% were obtained. The obtained results showed that no triazole fungicide residues were detected in all samples. These results confirmed that the proposed microextraction method could successfully be utilized to estimate triazole fungicide residues at trace levels in real samples with high accuracy and validity. The chromatograms of blank and spiked samples are shown in Figure 7.

### 2.5. Comparison of the Proposed Microextraction Method with Other Sample-Preparation Methods

Table 3 shows a comparison of the developed microextraction in this work with other published methods [1,4,29,30]. The ionic liquid could be prepared under facile conditions, which simplified the experimental operation, and no complicated instrument was required. Compared to the other sample-preparation methods, the proposed method provided a wide linear calibration range (150–1000 μg L^−1^), a high precision (less than 5), and acceptable recoveries (61–112) within shorter extraction times for simultaneous extraction determinations for various samples. In conclusion, the proposed microextraction method was demonstrated as a simple, fast, effective, and environmentally friendly technique.

## 3. Materials and Methods 

### 3.1. Chemicals and Reagents

Four triazole herbicides (myclobutanil (C_15_H_17_ClN_4_), triadimefon (C_14_H_16_ClN_3_O_2_), tebuconazole (C_16_H_22_ClN_3_O), and hexaconazole (C_14_H_17_C_l2_N_3_O)) from Dr. Ehrenstorfer GmbH (Augsburg, Germany) were used. The stock solution of each fungicide (1000 mg L^−1^) was prepared using methanol and used for further dilutions with water. Potassium hexafluorophosphate (KPF_6_) (Sigma-Aldrich, Beijing, China) and tributyldodecylphosphonium bromide ([P_44412_]Br) (Sigma-Aldrich, Schaffhausen, Switzerland) were used. Methanol and acetonitrile (HPLC-grade) were obtained from Merck (Darmstadt, Germany). Deionized water with a resistivity of 18.2 MΩ·cm was obtained from a Type 1 Simplicity^®^ ultrapure water system (Merck, Darmstadt, Germany). All solutions were filtered through a 0.45 μm filter before being injected into the HPLC system. All chemicals and reagents mentioned above were of analytical grade. 

### 3.2. Instrumentations

The HPLC analysis was conducted on a Waters 1525 Binary HPLC pump (Water, Massachusetts, USA) equipped with a diode array detector (DAD). A Rheodyne injector with a 20 µL injection loop was used. Empower 3 software was used to acquire and analyze the chromatographic data. A Purospher^®^ STAR RP-18 endcapped (4.6 × 150 mm, 5 µm) column (Merck, Darmstadt, Germany) with an isocratic elution of acetonitrile and water at a ratio of 50:50 (%*v*/*v*) was used for the separations. The mobile phase flow rate was 1 mL min^−1^. The detection wavelength was set at 220 nm. 

The Fourier-transformed infrared spectra (FTIR) of the ionic liquid phase were measured using a Bruker Invenio-S FT-IR (Bruker Corp., Massachusetts, USA). Diamond-lens-attenuated total reflectance (ATR) was used. A centrifuge (Centurion, Aberdeen, England) also was used. 

### 3.3. Sample Preparation

#### 3.3.1. Honey Samples

Honey samples were purchased from a supermarket in Maha Sarakham province. A total of 5 g of the sample was weighed into a 50 mL volumetric flask and diluted to the marker. The solution was filtered through a Whatman (no. 1) filter paper. After that, the filtrate was passed through a 0.45 μm nylon membrane filter before extraction using the proposed method. 

#### 3.3.2. Fruit Juice Samples 

Passion fruit juice and pomegranate juice (commercial juice samples) were bought from the supermarket in Maha Sarakham province. An aliquot of fruit juice (30.0 mL) was centrifuged at 4000 rpm for 10 min and then filtered through a Whatman (no. 1) filter paper. The solution was then passed through a 0.45 μm nylon membrane filter before extraction using the proposed method.

#### 3.3.3. Egg Yolk Sample

Chicken eggs were purchased from local markets in Maha Sarakham province. The yolk was separated from the white to reduce interference, since in the analysis of egg collected from animals treated with anthelmintics, it is known that the interferences are greater in the yolk [31,32]. Fortification of the sample was performed directly in the yolk, and a period of about 12 h was allowed to elapse before continuing with any of the extraction processes in order to improve the interaction between the analytes and the matrix compounds [33]. A total of 10.00 g of egg yolk was mixed well with 0.2 g of anhydrous Na_2_SO_4_. After that, 1% (*v/v*) acetic acid in acetonitrile (2.00 mL) was added and shaken vigorously by hand for 1 min, and the homogenized eggs were centrifuged at 3500 rpm for 5 min for complete fat and protein precipitation. The supernatants were collected using a microsyringe. The solutions were diluted with deionized water to 10.00 mL, 100 μL of acetic acid was added, and the solutions were centrifuged to ensure the complete precipitation of fat and proteins [32]. The samples were spiked with the triazole fungicides at different concentrations before fat and protein precipitation. The obtained clear solutions were then extracted using the proposed microextraction method.

### 3.4. In Situ Metathesis-Reaction-Generated Ionic Liquid Combined with Liquid–Liquid Microextraction

A schematic diagram of the microextraction procedure is shown in Figure 8. The mixed solution included the standard or sample solution (10.00 mL), 0.01 g of [P_44412_]Br and 0.01 g of KPF_6_, which were added into a 15 mL conical centrifuge tube. Then, the tube was shaken to dissolve the [P_44412_]Br and KPF_6_ and to complete the in situ metathesis reaction. The solution was then centrifuged at 4500 rpm for 10 min. After that, the centrifuge tube was cooled in an ice bath until the [P_44412_]PF_6_ was generated. The [P_44412_]PF_6_ phase (upper phase) was separated and then diluted with acetonitrile (200 μL) to decrease the viscosity before being injected into the HPLC system.

### 3.5. Evaluation of Enrichment Factor (EF), Extraction Recovery (ER), Relative Recovery (RR), and Matrix Effect (ME)

To study the effects of experimental conditions on the extraction efficiency, the EF was calculated between the analyte concentration in the final phase (C_final_) and the initial concentration in the analyte in aqueous sample solution (C_0_) according to the following equations:EF = C_final_/C_0_(2)

The %ER was expressed as the total percentage amount of the target analytes extracted into the sediment phase using the proposed microextraction method:%ER = EF × (V_sed_/V_0_) × 100(3)
where V_sed_ and V_0_ are the volume of sediment phase and the sample solution, respectively.

The %RR was defined as the % amount of analyte recovered from the matrix (real samples) with reference to the extracted standard (standard spiked into the same matrix).
(4)$$\mathrm{RR}\;(\%)=\frac{C_{\mathrm{found}}-C_{\mathrm{real}}}{C_{\mathrm{added}}}\times 100$$
where C_found_ is the concentration of analyte after adding a known amount of working standard to the real sample, C_real_ is the analyte concentration in the real sample, and C_added_ represents the concentration of a known amount of working standard that was spiked into the real samples.

## 4. Conclusions

In the present study, an in situ extraction and preconcentration method using an ionic liquid for separation and preconcentration of triazole fungicides in honey, fruit juice, and egg samples was performed prior to high-performance liquid chromatographic analysis. The halide anion of tributylhexadecylphosphonium bromide ([P_44412_]Br) and potassium hexafluorophosphate (KPF_6_) was used for the formation of the hydrophobic ionic liquid. In the proposed microextraction method, forming the immiscible IL extraction phase and the transfer of analytes occurred simultaneously. The metathesis reaction and extraction were accomplished in single step, making the transfer of the analytes into the extracting phase very quick and efficient. The proposed method provided good repeatability, a wide linearity range, a high enrichment factor, and an acceptable extraction recovery for each compound, and matrix effects did not interfere with the quantification process. Therefore, the proposed method is recommended as a fast, simple, sensitive, and environmentally friendly sample-preparation technique.

## Figures and Tables

**Figure 1 molecules-27-03416-f001:**
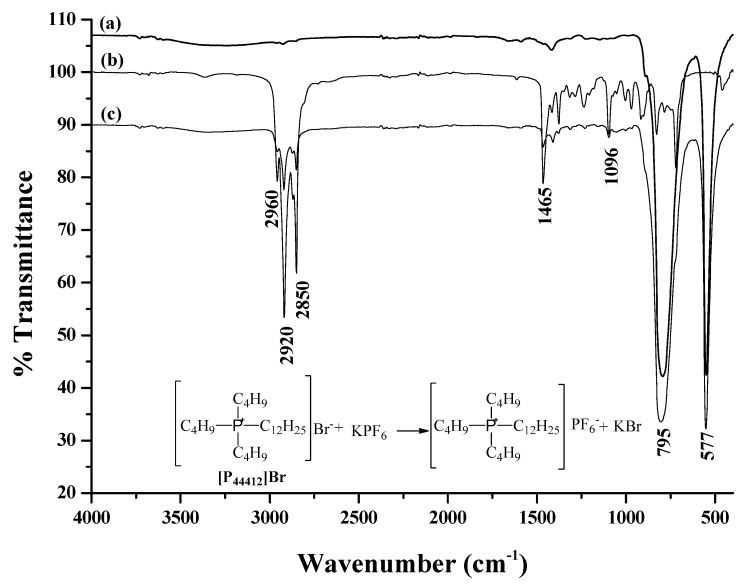
FT–IR spectra of (**a**) KPF_6_, (**b**) [P_44412_]Br, and (**c**) the in situ ionic liquid when it was formed.

**Figure 2 molecules-27-03416-f002:**
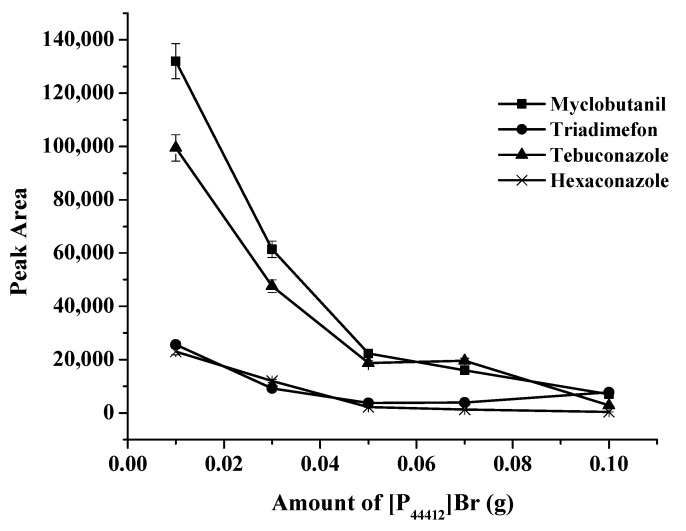
Effects of the amount of [P_44412_]Br on the extraction efficiency.

**Figure 3 molecules-27-03416-f003:**
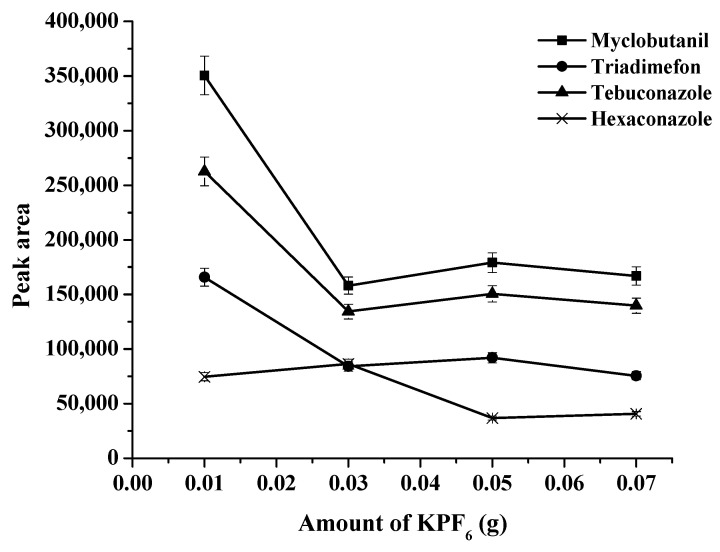
Effects of the amount of KPF_6_ on the extraction efficiency.

**Figure 4 molecules-27-03416-f004:**
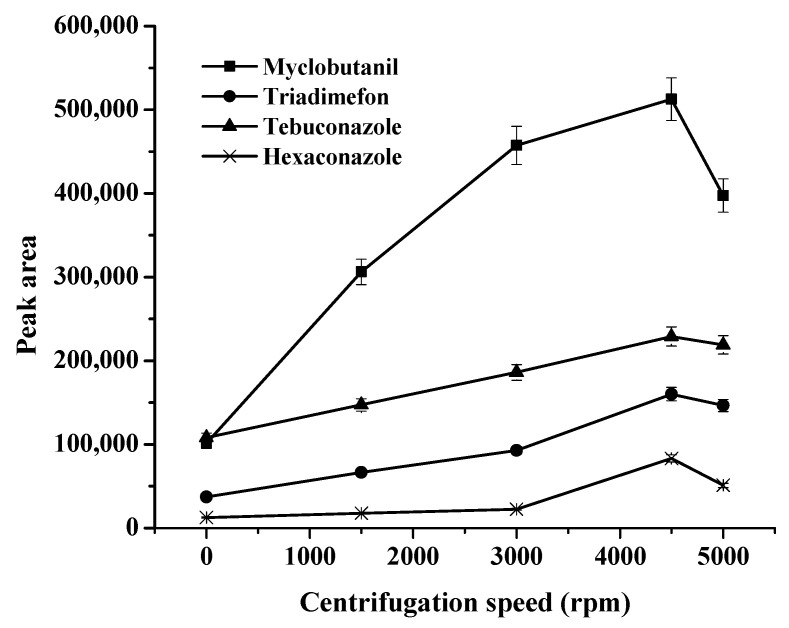
Effects of centrifugation speed (rpm) on the extraction efficiency.

**Figure 5 molecules-27-03416-f005:**
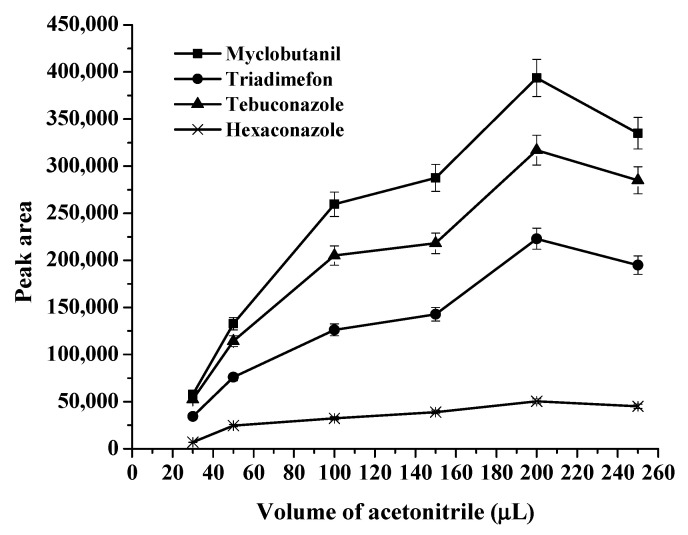
Effects of dissolving solvent (acetonitrile, μL) on the extraction efficiency.

**Figure 6 molecules-27-03416-f006:**
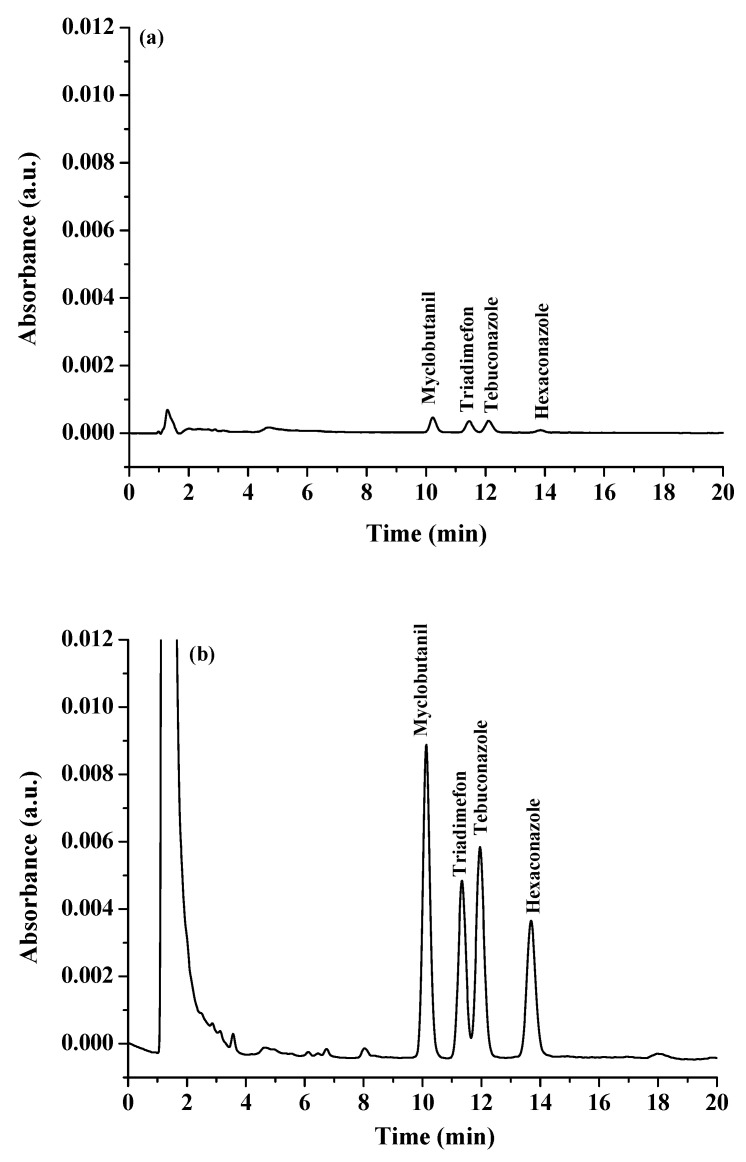
Chromatograms of standard neonicotinoid insecticides obtained (**a**) without preconcentration and (**b**) with preconcentration using the proposed microextraction method; the concentration of all standards was 300 µg L^−1^.

**Figure 7 molecules-27-03416-f007:**
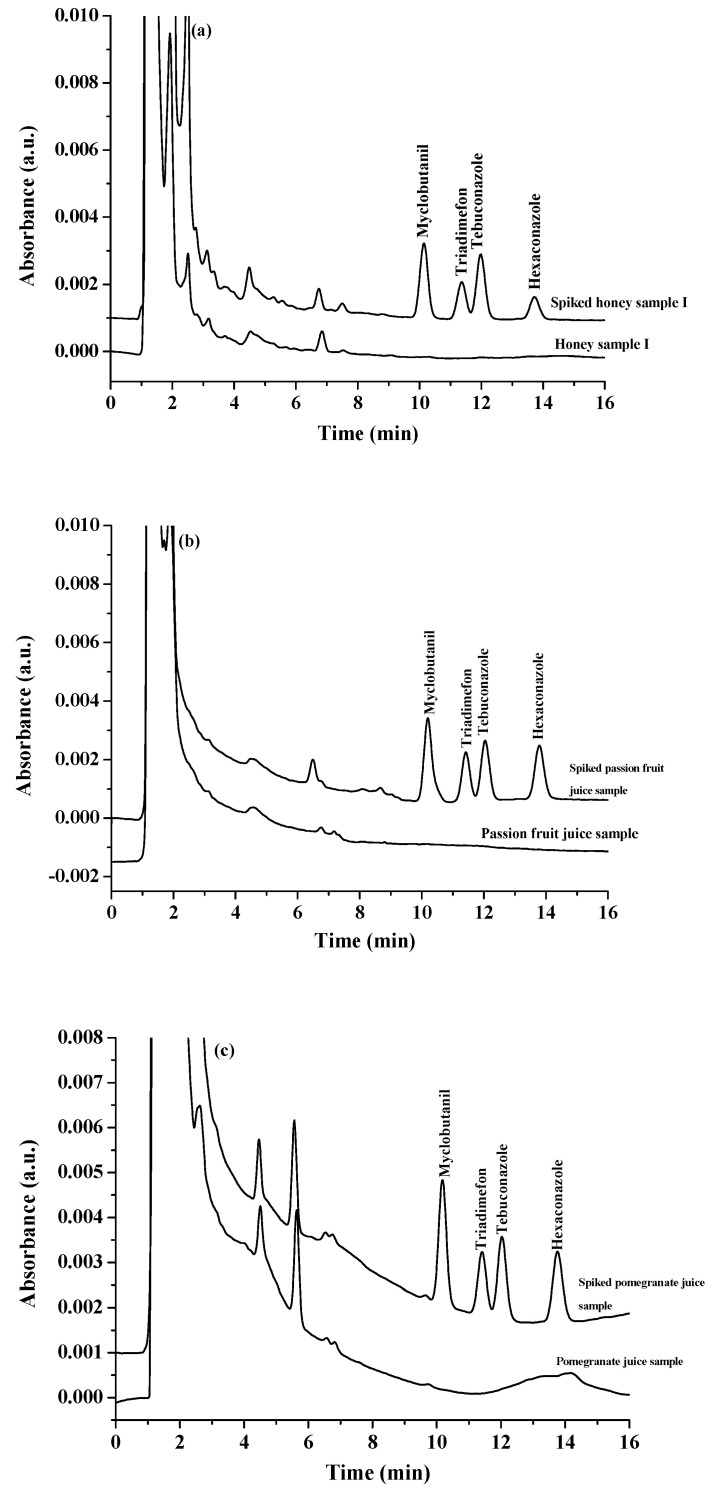
Chromatograms of sample and spiked sample (150 μg L^−1^ for fruit juice sample and 150 μg kg^−1^ for honey and egg samples); (**a**) honey sample, (**b**) passion fruit juice sample, (**c**) pomegranate juice sample and (**d**) egg yolk sample.

**Figure 8 molecules-27-03416-f008:**
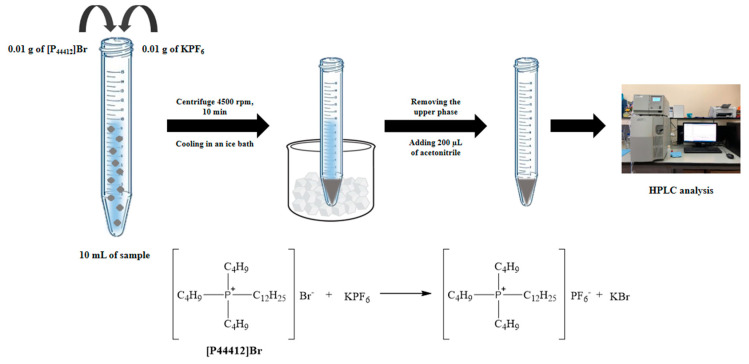
Schematic illustration of the microextraction procedure using in situ formation of the ionic liquid for triazole fungicides and HPLC analysis.

**Table 1 molecules-27-03416-t001:** Analytical performance of the proposed method.

Analyte	Linear Range (µg L^−1^)	Linear Equation	R^2^	LOD (µg L^−1^)	LOQ (µg L^−1^)	Intraday Precision (n = 5), RSD (%)		Interday Precision (n = 5), RSD (%)		EF	ER
						t_R_	Peak Area	t_R_	Peak Area		
Myclobutanil	90–1000	y = 1 × 10^6^x + 9995.5	0.9994	30	90	0.61	2.48	0.77	2.70	12.26	85.82
Triadimefon	90–1000	y = 907,260x + 11,705	0.9989	30	90	0.51	2.92	0.80	3.50	8.53	59.71
Tebuconazole	90–1000	y = 484,103x + 4328.4	0.9991	30	90	0.40	2.60	0.46	2.72	11.01	77.07
Hexaconazole	150–1000	y = 689,281x + 3751.6	0.9988	50	150	0.70	2.55	0.77	2.60	11.34	79.38

**Table 2 molecules-27-03416-t002:** Recoveries obtained from the analysis of triazole fungicides in real samples (n = 3).

Sample	MCBT		TDF		TBZ		HCZ	
	%RR *	%RSD **	%RR	%RSD	%RR	%RSD	%RR	%RSD
Honey I	66.17	0.94	92.41	2.82	99.72	1.99	94.12	1.06
Honey II	86.98	2.47	72.03	3.12	60.26	2.60	62.22	2.57
Passion fruit juice	76.14	0.84	77.87	1.80	81.50	1.82	87.63	1.20
Pomegranate juice	72.14	0.71	61.77	1.78	69.19	0.20	62.98	1.53
Grape juice	72.50	1.82	62.65	1.39	66.16	0.87	63.59	2.87
Egg yolk	84.19	0.31	72.72	0.47	69.41	1.69	69.05	1.08

* RR: relative recovery; ** RSD: relative standard deviation.

**Table 3 molecules-27-03416-t003:** Comparisons of the proposed method with other methods for the quantitation of triazole fungicides.

Extraction Method **	Analytical Method	Samples	Adsorbent/Desorbed ***	Linear Range (μg L^−1^)	Recovery (%)	LOD (μg L^−1^ or μg Kg^−1^)	EF	%RSD	Ref.
SMMH–d–SPME	HPLC–UV	Water	Fe_3_O_4_@SiO_2_/ACN	5–100 and 2.5–50	90–104	1.0–2.5	40–237	Less than 8%	[4]
MSPE	HPLC–UV	River water	BCDP/ACN – 0.1% (*v*/*v*) HCOOH	1–1200	82.8–113.2	0.2–0.3	281–283	1.2–4.6	[29]
SBSE	HPLC–DAD	Grape and cabbage	HC–POF/ACN–water	0.1–500	80.7–111	0.022–0.071	49–57	6.4–12.4	[1]
MSPE	LC–MS/MS	Water and fruit juices	GO–P*m*AP/ACN–water in NaOH	0.001–0.5	80.3–106.3	0.08–2.04 ng g^−1^	0.45–0.60	2.1–13.4	[30]
Ionic liquid combined with liquid–liquid microextraction	HPLC–DAD	Honey, fruit juices, and egg yolk	IL–ACN	150–1000	61–112	30–50	8–11	≤5	This work

** MSPE-HPLC-UV: magnetic solid-phase extraction–high-performance liquid chromatography–ultraviolet detector; SBSE-HPLC-DAD: stir bar sorption extraction–high-performance liquid chromatography–diode array detector; SPME-HPLC-DAD: solid-phase microextraction–high-performance liquid chromatography–diode array detector; MSPE-LC-MS/MS: magnetic solid-phase extraction–liquid chromatography–tandem mass spectrometry; µSPE-HPLC-PDA: micro solid-phase extraction–high-performance liquid chromatography–photodiode array; SMMH-d-SPME: in situ surfactant-mixed metal hydroxide (SMMH); d-SPME: dispersive solid-phase microextraction; *** Fe_3_O_4_@SiO_2_: magnetic nanoparticle; ACN: acetonitrile; BCDP: bridged *bis*CD-bonded chiral column; HCOOH: formic acid; HC-POF: hydroxyl-containing porous organic framework; GO-P*m*AP: graphene oxide-poly 3-aminophenol.

## Data Availability

The data presented in this study are available on request from the corresponding author.

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
