# Peer review of "An In Situ Formation of Ionic Liquid for Enrichment of Triazole Fungicides in Food Applications Followed by HPLC Determination"

_molecules, 2022, doi:10.3390/molecules27113416_

Round 1
Reviewer 1 Report
I have read the manuscript “An in-situ formation of ionic liquid coupled to high performance liquid chromatography for preconcentration of triazole fungicides in food samples” by Rawikan Kachangoon et al. (MS # molecules-1731452) submitted for the publication in Molecules.
In their manuscript the authors investigated the use of an in-situ ionic liquid for the extraction of pollutants (triazole fungicides) and applied the optimized procedure for the determination of such fungicides from real food matrices.
The topic is obviously interesting and timely, but, it is opinion of the referee, that the manuscript needs revisions before its publication in Molecules.
In particular:
- English needs revisions. There are several misprints (e.g. lines 38, 39, 102, 109, 159, 177, 216) including the name of a fungicide (hexzaconazole). Acronyms must be defined the first time they are cited (e.g. in Figures 2, 3, 4, and 5).
- The authors should report the rationale for using the chosen ionic liquid (two components in-situ forming) and the advantages with respect others (one component ready to use).
- Line 112: The ionic liquid properties should be better defined and reported.
- Line 177: for an easy reading the equation should be reported.
- Line 187 and caption of Figure 8: the concentration values of fungicides are different.
- Line 205: the authors should discuss the relatively higher values for their linear ranges and LODs than those found in literature.
- Line 239: reflectance rather than resistance.
- Equation 1: I suppose that Csed was obtained by difference between C0 and Cfinal.
- Some references are doubled numbered.
Author Response
We thank you and the reviewers very much for the valuable comments. The manuscript has been carefully revised accordingly to the comments as in the following: Manuscript entitled “An in-situ formation of ionic liquid coupled to high performance liquid chromatography for preconcentration of triazole fungicides in food samples”
All changes are highlighted in the revised manuscript using red colored.
Thank you very much for consideration of our manuscript.
Yours sincerely,
Authors

Reviewer 2 Report
Minor changes:
Line 38: “for a” instead of “for a”
Line 95: the shift to 832 cm-1 can not be seen in Fig 1. Please, clarify
Line 109: “need to be investigated”
Line 132: the sentence starting with “when we used over that speed” should be rewritten for better understanding
Line 145 and Figure 5: Please explain the effect of acetonitrile volume
Figure 6: is the signal voltage or absorbance? The same in Figure 8
Table 3: the second reference cited should be 29 and not 39
References: Please check numbering and avoid duplication
Author Response

(The authors gave the same response as above.)
